# Na⁺-Taurocholate Co-Transporting Polypeptide (NTCP) in Livers, Function, Expression Regulation, and Potential in Hepatitis B Treatment

**Xiaoyu Zhao** [1,2,3], **Waqas Iqbal** [1,2,3], **Pingnan Sun** [1,2,3] **and Xiaoling Zhou** [1,2,3,*]

1   Stem Cell Research Center, Shantou University Medical College, Shantou 515041, China;
    1155160436@link.cuhk.edu.hk (X.Z.); waqasbiotech@yahoo.com (W.I.); pnsun@stu.edu.cn (P.S.)
2   Research Center for Reproductive Medicine, Shantou University Medical College, Shantou 515041, China
3   Guangdong Provincial Key Laboratory of Infectious Diseases and Molecular Immunopathology,
    Shantou University Medical College, Shantou 515041, China
*   Correspondence: xlzhou@stu.edu.cn

**Abstract:** Chronic hepatitis B virus (HBV) infection has become one of the leading causes of liver cirrhosis and hepatocellular carcinoma globally. The discovery of sodium taurocholate co-transporting polypeptide (NTCP), a solute carrier, as a key receptor for HBV and hepatitis D virus (HDV) has opened new avenues for HBV treatment. Additionally, it has led researchers to generate hepatoma cell lines (including HepG2-NTCP and Huh-7-NTCP) susceptible to HBV infection in vitro, hence, paving the way to develop and efficiently screen new and novel anti-HBV drugs. This review summarizes the history, function and critical findings regarding NTCP as a viral receptor for HBV/HDV, and it also discusses recently developed drugs targeting NTCP.

**Keywords:** NTCP; HBV; anti-HBV drugs



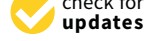

## 1. Introduction

Chronic hepatitis B virus (HBV) infection remains a health issue globally. According to the World Hepatitis Summit in 2017, more than 240 million people suffer from chronic hepatitis B or C infection globally. At present, the World Health Organization has listed viral hepatitis as one of the major threats to global public health. Although vaccination has proven successful, the high turnover rate has resulted in vaccine-related escape viral mutants, making the HBV infection a serious problem [1]. Currently available conventional therapies are based on PEGylated-interferon and nucleoside analogs. Interferon (IFN) as an immunomodulator directly inhibits HBV replication by decreasing the transcription of an intermediate replicative RNA known as pregenomic RNA (pgRNA), in addition to subgenomic RNA from the HBV covalently closed circular DNA (cccDNA). Nucleoside analogs inhibit HBV replication mainly through inhibiting the reverse transcription process of the viral lifecycle. Although these two kinds of drugs are effective in inhibiting HBV replication, their side effects cannot be underestimated. Long-term usage of these antiviral agents may reduce patient compliance and increase drug resistance [2,3]. Therefore, it is necessary to find new therapeutic approaches to HBV cure by targeting different factors related to HBV infection and spread.

HBV is a hepatotropic DNA virus. The HBV virion, also called the Dane particle, consists of a 3.2 kb partially double-stranded and circular DNA genome encoding four overlapping reading frames. Despite its small size, the genome is capable of encoding proteins necessary for complete viral replication. Hepatitis delta virus (HDV) is a small satellite RNA virus that uses the envelope of HBV to complete its lifecycle; HBV and HDV can be treated with the same drugs.

Na$^+$-taurocholate co-transporting polypeptide (NTCP) belongs to the solute carrier family 10 members (SLC10) and is encoded by the SLC10A1 gene. The SLC10, a transporter gene family, contains seven members out of which three (SLC10A1, SLC10A2, and SLC10A6) are Na$^+$-dependent co-transporters. SLC10A1 and SLC10A2 (apical Na$^+$-dependent bile salt transporter (ASBT)) transport bile salts and maintain enterohepatic circulation of bile salts [4].

Besides transport function, NTCP also mediates HBV and HDV entry into hepatocytes [5]. This momentous discovery also accelerated the development of anti-HBV/HDV drugs. However, the precise role of NTCP in viral entry is not clear [6]. Nevertheless, binding of HBV preS1 region blocks taurocholate entry, on the other hand, certain bile acid substrates could potentially block HBV entry, making them potential candidates for antiviral drug discovery [7].

Moreover, NTCP can also influence the hepatitis C virus (HCV) infection process, where NTCP modulates HCV infection through bile acid-mediated suppression of interferon-stimulated genes (ISGs) [8].

Hence, there has been a surge in research focusing on NTCP as a drug target to inhibit HBV and HCV infections. This review summarizes the research history, functions, expression, and drug development of NTCP. It also sheds light on anti-HBV drugs that specifically target NTCP.

## 2. History of NTCP Research

Na$^+$-dependent bile acid transport was first observed in rat hepatocytes in 1978 [4]. The first rNtcp and hNTCP orthologue were cloned in 1991 and 1994, respectively [9]. rNtcp is a seven-transmembrane-spanning protein that is localized at the basolateral plasma membrane of hepatocytes [10], whereas ASBT from *Neisseria meningitides* (ASBT$_{NM}$) contains ten transmembrane helices [11]. hNTCP is purported to contain seven to nine transmembrane domains [12]. Na$^+$-dependent human bile acid transporters of the SLC10 family (NTCP and ASBT), which are expressed in hepatocytes and intestinal epithelial cells, respectively, are electrogenic [13]. A number of factors, such as liver-enriched transcription factors, drugs including dexamethasone, hormones and proinflammatory cytokines such as, IL-1β and IL-6, have been found to regulate Ntcp/NTCP expression [14–18], additionally, cyclosporine regulates NTCP's transport activity [19]. Some FDA-approved NTCP targeting HBV entry inhibitors have been identified [20,21]. A milestone in NTCP research was the discovery of HBV/HDV receptor. In 2012, Li et al. applied near zero distance photo-cross-linking and tandem affinity purification to identify hNTCP as a functional receptor for both HBV and HDV. They showed that hNTCP specifically binds to HBV envelope protein (preS1) and mediates viral infection [5]. Discoveries related to Ntcp/NTCP have been summarized in Table 1.

**Table 1.** Discoveries in NTCP.

| Years | Discoveries | References |
|---|---|---|
| 1978 | Bile salt transport in rat hepatocytes is Na$^+$-dependent | [4] |
| 1991 | The first rNtcp orthologue which contains seven transmembrane spanning domain was cloned | [10] |
| 1991 | Rat Ntcp has seven-transmembrane-spanning domains and five putative N-linked glycosylation sites | [10] |
| 1993 | Na$^+$-dependent taurocholate uptake is inhibited by cyclosporine in human hepatocytes | [22] |
| 1994 | The first human NTCP orthologue was cloned | [9] |
| 1994 | NTCP is encoded by the SLC10A1 gene in humans | [9] |
| 1996 | human HepG2 is lack of NTCP expression | [23] |
| 1996 | Ntcp transports two sodium ions together with one bile salt molecule | [24] |
| 1997 | NTCP is localized on the basolateral plasma membrane of human hepatocytes | [25] |
| 1997 | NTCP is an electrogenic transporter | [13] |

**Table 1.** *Cont.*

| Years | Discoveries | References |
|---|---|---|
| 2001 | A free C-terminal part of human NTCP is not essential for function | [26] |
| 2002 | Expression of Ntcp is observed in pancreatic acinar cells | [27] |
| 2003 | Expression of NTCP is observed in placenta which may explain maternal-neonatal HBV transmission | [28] |
| 2004 | The process of NTCP transport bile salt uptake from portal blood into liver is Na$^+$ dependent | [29] |
| 2009 | NTCP extracts the majority of conjugated bile acids at the basolateral membrane of the liver | [30] |
| 2012 | NTCP was discovered to be the primary receptor for HBV entry | [5] |
| 2013 | Thirty-one FDA-approved drugs were screened for inhibition of NTCP-dependent transport. | [21] |
| 2014 | HBV entry and bile salt transport share common molecular determinants in NTCP | [7] |
| 2015 | IL-6 blocks HBV entry by downregulating NTCP | [16] |
| 2016 | Cyclosporine derivatives inhibit HBV entry without interfering with NTCP transporter activity | [31] |
| 2016 | Huh-7-NTCP cells can produce more infectious HBV pseudoparticles than parental Huh-7 cells | [32] |
| 2017 | N-Glycosylation of NTCP is essential for HBV infection | [33] |
| 2017 | NTCP forms a stable bile acid uptake machinery in humans | [34] |

## 3. Functions of NTCP

### 3.1. NTCP as a Transporter for Bile Salts

NTCP is a co-transporter of Na$^+$ and bile acids in hepatocytes with a Na$^+$/taurocholate stoichiometry of 2:1. In addition to the primary function of extracting bile salts, such as taurocholic acid, taurodeoxycholic acid, and glycocholic acid from sinusoidal blood, NTCP also transports conjugated sex hormones, thyroid hormones, bromosulfophthalein, and even exogenous chemicals [35]

The bile is synthesized in the liver and stored in the gallbladder, and plays an important role in intestinal absorption, digestion of lipids, and cholesterol metabolism. After stimulation with foods, it is excreted into the duodenum and small intestine. At the end of the ileum, most of the bile is absorbed by the intestinal wall and returned to the liver through the portal vein and re-secretion accordingly [36].

Bile acid from sinusoidal blood enters canaliculus through the hepatocytes (Figure 1). Bile acids are taken up by hepatocytes via NTCP. Bile acids flow out the basolateral membrane of hepatocytes by the multidrug resistance proteins MRP3 and MRP4. Bile acids are secreted across the canaliculus membrane by two members of the ABC family: the bile acids export pump (BSEP) [37] and multidrug resistance protein 2 (MRP2) [38].

### 3.2. NTCP as a Functional Receptor for HBV/HDV

The key determinant for entry of HBV/HDV is the preS1 domain of the L protein expressed on the HBV/HDV surface. The functional receptor on hepatocytes remained unknown for more than twenty years. Recently, hNTCP was identified to be critical for preS1 binding and viral infection.

Interestingly, NTCP is broadly expressed in the liver of all species and presents a high amino acid conservation (across 66.2%) among mammals [39], but HBV infection is found exclusively in humans, chimpanzees, and Tupaias. Woodchuck NTCP also supports HBV and HDV infection, but at low levels. In addition, laboratory animals, such as mice, rats, and macaques are resistant to HBV and HDV infection. These demonstrate that HBV has a limited host range, compounding in vivo studies. The reasons are closely related to NTCP variation, expression level, or lack of a system for internalizing HBV after the NTCP attachment step.

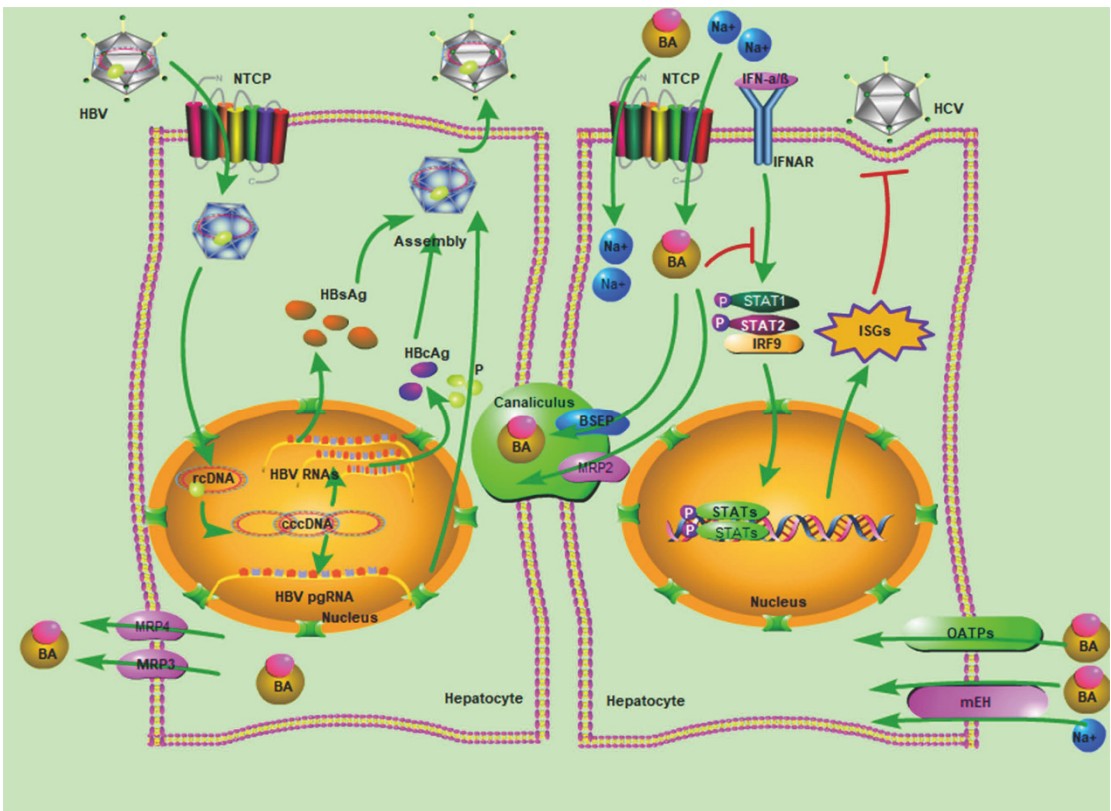

**Figure 1.** Multiple functions of NTCP. Initially, bile acids (BAs) are transported by NTCP in hepatocytes. This figure shows the process of bile acid transport. Secondly, HBV enters cells via NTCP. The entire lifecycle of HBV includes attachment, entry, uncoating and nuclear import, cccDNA formation, transcription, translation, encapsidation, replication, assembly, and secretion. In the left cell, we show the complete lifecycle of HBV. Thirdly, the bile acids transported from the extracellular environment via NTCP inhibit the expression of ISGs, thereby inhibiting HCV infection. NTCP, Na$^+$-taurocholate co-transporting polypeptide; BSEP, bile salt export pump; MRP4, multidrug resistance-associated protein 4; MRP3, multidrug resistance-associated protein 3; MRP2, multidrug resistance-associated protein 2; OATPs, organic-anion-transporting polypeptide; mEH, microsomal epoxide hydrolase; ASBT, apical sodium-dependent bile salt transporter.

With the discovery of NTCP as a key receptor of HBV infection, hNTCP residues 157 to 165 (KGIVISLVL) were identified to play an important role in this process [5]. By analyzing the susceptibility of five NTCP variants (S142T, K157G, G158R, V160I, and L165P), Simon et.al have shown that amino acid position 158 of NTCP/Ntcp is critical for HBV/ HDV infection, which 158G (glycine) is susceptible whereas 158R (Arginine) is non-susceptible [40]. However, G158R also decreases the taurocholate uptake by nearly 60%. Similarly, the substitution of serine at position 267 of NTCP with phenylalanine (S267F) [41] and the substitution of alanine at position 64 of NTCP with threonine (A64T) [42] previously were found to cause substantial loss of ability, and supported HBV and HDV infection and its bile acid uptake capability in vivo.

There are differences in allele frequencies of NTCP variants, such as race-specific variant S267F. It loses the function of transporting bile salts and the capability of binding HBV surface protein preS1, and thus blocks HBV entry. The allele frequencies of NTCP S267F ethnically vary by 2.4–8.3% in Chinese [43], 7.5% in Chinese-Americans [44], 9.2% in Vietnamese, and 3.1% in Koreans [42], while it was not found in European-Americans and African-Americans [44]. It was suspected that a higher frequency of variant S267F in the Asian population might have resulted from NTCP gene evolution under a higher frequency of HBV infection [45]. However, a controversial relationship still remains between S267F polymorphisms and HBV susceptibility [46–48].

Recently, Urban's group found that cynomolgus, rhesus macaques, and pig primary hepatocytes over-expressing hNTCP become fully susceptible to both HDV and HBV [49]. In vivo, Benjamin's group also confirmed HBV infection in hNTCP-expressing rhesus macaques [50]. However, hNTCP-expressing hepatocytes of mice, rats, and dogs could initiate HBV attachment but not support HBV infection. It was supposed that other host factors besides NTCP were also required for HBV internalization in hepatocytes of these animals [44].

In in vitro cell models, the HepaRG cell line [51], primary human hepatocytes (PHHs) [52], and primary Tupaia hepatocytes were found to be susceptible to HBV, whereas HepG2 and Huh-7 cell lines did not get infected due to low NTCP expression [25]. In contrast to HepG2 or Huh-7, the HepaRG cell line maintains a high degree of physiological hepatic function and demonstrates a transcriptomic pattern more closely resembling that of hepatocytes. However, HepaRG is restrictive, since it requires a long-term differentiation process that may affect the reproducibility of experiments, and, in addition, the infection efficiency is low. PHHs remain the gold standard, however, the ability of PHHs to be infected by HBV decreases rapidly after plating because of the loss of hepatocyte polarization under culture conditions [52]. Additionally, genetic variations among donors also make studies' reproducibility difficult. Moreover, limited availability and rapid dedifferentiation in vitro make PHHs less desirable. Therefore, HepG2 expressing hNTCP (HepG2-hNTCP) is now widely used as a novel infection model to study HBV/HDV infection and to screen anti-HBV drugs.

The HepG2-NTCP and Huh-7-NTCP cell lines are efficient HBV infection models, and easy to culture in vitro due to their cancer cell characteristics. These two cell lines can effectively recapitulate HBV infection processes in vitro. Nevertheless, HepG2 and Huh-7 cells are hepatoma cells with aberrant gene expression, long-term culturing results abnormal chromosomal copy numbers, and disrupted epigenetic states, hence they cannot fully reflect actual virus-host interactions.

Recently, umbilical cord matrix stem cells differentiated into hepatocyte-like cells, resulting in susceptibility to HBV infection, were used to study the early stages of viral entry by endogenous hNTCP [53]. More importantly, human stem cell-derived hepatocyte-like cells (HLCs), which mimic characteristics of PHH better than other cell models, are permissive to and support productive HBV infection [54]. In the future, HLCs might be used more frequently as a good model for screening anti-HBV drugs, with the aim of targeting NTCP, since HLCs have endogenous NTCP expression and closely resemble PHHs. Additionally, HLCs can be maintained for a longer period of time in vitro as compared to PHHs.

In summary, HBV is not susceptible to all hNTCP-expressing hepatocytes, and hNTCP level and HBV infection rate may be not in parallel. NTCP expression level of HepaRG-NTCP cells was higher than that of HepaRG cells, and HBV infection rate of HepaRG-NTCP cells (~40%) was also higher than HepaRG cells (~20%). However, stem cell-derived hepatocytes expressed a higher level of NTCP than human primary hepatocytes but the former had a lower HBV infection rate than the latter [55]. In addition, Koichi Watashi's group also showed that different HepG2-NTCP clones with similar NTCP expression levels had diverse efficiencies of HBV infection [6,56]

### 3.3. NTCP and HCV Infection

HCV belongs to the family Flaviviridae and is a single-stranded positive-strand RNA virus. HCV has a full-length genome of about 9.6 kb that encodes a core protein (C protein) and structural proteins, membrane glycoprotein E1 and E2, as well as one viroprotein P7 and non-structural protein.

HCV can block innate immune signaling on several levels, yet induces a strong IFN response in PHHs, chimpanzees [57], and acutely infected patients [57]. Recently, NTCP was discovered to interfere with HCV infection by modulating IFN signaling pathway in PHHs, and NTCP overexpression enhances HCV infection whereas silencing NTCP

expression inhibits HCV infection [8]. However, HCV and HBV interact differently with NTCP (Figure 1). Previously, it was discovered that bile acid has an inhibitory effect on the ISGs, such as antiviral proteins MX1 and OAS1 in the interferon signaling pathway [58]. The HCV structural protein E2 does not bind to NTCP, in fact bile acids transported via NTCP suppress the expression of antiviral ISGs, thus resulting in increased HCV infectivity.

## 4. Regulation of NTCP Expression

The hNTCP gene spans 21.4 kb and maps to chromosome 6q24. It encodes 349 amino acids and the calculated molecular mass of the protein is approximately 38 kD. The gene comprises of five exons and contains an open reading frame of 1047 bp. Substrates, cytokines, hormones, diseases, and even physiological factors can regulate the expression of NTCP.

### 4.1. NTCP Expression under Physiological and Pathological Conditions

Similar NTCP expression patterns have been observed in human and rodents. The expression is almost 50-fold higher in adult livers compared to fetal livers [59]. The expression of rNtcp has been detected at approximately 20 days of gestation, reaching expression levels found in adults at postnatal day 28 [60]. Mouse NTCP (mNtcp) mRNA expression in liver rapidly increases to the highest level at birth, but it decreases after birth, and then increases to adult levels at approximately 3 weeks of age [61].

Gender-dependent expression of NTCP is observed among rats, mice, and humans. In rats, rNtcp mRNA expression is male-predominant [14]. The gender-based differences in NTCP mRNA levels result from inhibitory effects of female-pattern growth hormone (GH) secretion. Conversely, thyroid hormone, glucocorticoids, and corticosterone may increase rNtcp mRNA expression [14]. Interestingly, mNtcp mRNA expression is female-predominant due to inhibitory effects of male-pattern GH secretion, but not sex hormones. In humans, hNTCP mRNA levels are comparatively higher in women than in men, but without statistical difference [61].

With regard to disease, NTCP expression has been shown to be downregulated under pathological conditions, such as progressive familial intrahepatic cholestasis, inflammatory cholestasis, primary biliary cirrhosis, cholestatic alcoholic hepatitis, and chronic hepatitis C, while NTCP is upregulated in nonalcoholic steatohepatitis, end-stage primary biliary cholangitis, and in patients with late-stage obstructive cholestasis. The NTCP variant (S267F) is independently associated with decreased risk of cirrhosis and hepatocellular carcinoma (HCC), and resistance to chronic hepatitis B infection [62]. Additionally, the hNTCP mRNA expression and protein levels are significantly decreased in patients with HCC. Moreover, lower NTCP expression is associated with poor prognosis and lower HBV cccDNA levels in HCC patients [63].

### 4.2. Factors Regulating NTCP Expression

With regard to the substrate feedback, hNTCP expression was downregulated by high levels of bile acids as an adaptive response to block excessive bile acid accumulation in hepatocytes [64]. Bile acids induce the inhibition of NTCP transcription via activation of the farnesoid X receptor (FXR), a bile acid-activated nuclear receptor. FXR indirectly affects NTCP transport activity, although it does not interact with NTCP promoter [64]. In hepatocytes, activated FXR promotes the expression of short heterodimer partner (SHP), which in turn blocks the stimulating effect of retinoic acid receptor and retinoid X receptor heterodimers (RAR/RXRs) on the NTCP promoter (Figure 2).

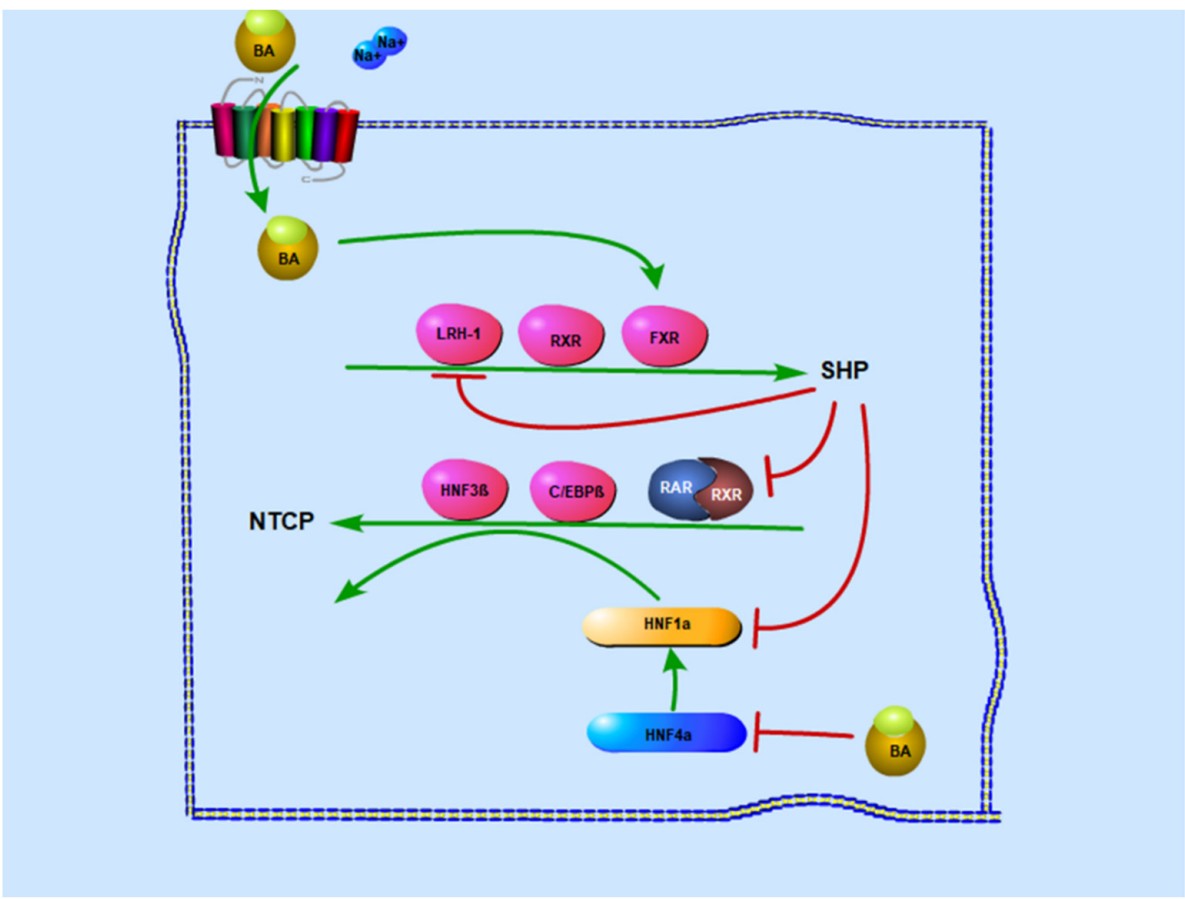

**Figure 2.** Regulation of NTCP expression levels in vivo. There are mainly two pathways to affect the NTCP expression levels in vivo involving bile acid. On the one hand, bile acids stimulate SHP expression by activating FXR response elements in the SHP promoters, whereas SHP decreases NTCP expression by interfering with RXR: RAR transactivation on the NTCP promoter. At the same time, SHP also inhibits HNF4α-mediated transactivation of the HNF1α promoter, to further inhibit NTCP expression. On the other hand, intracellular bile acid also decreases NTCP expression by suppressing HNF1α via the inhibition of HNF4α.

With regard to liver-enriched transcription factors and nuclear receptors, hepatocyte nuclear factor 1 alpha (HNF1α), hepatocyte nuclear factor 4 alpha (HNF4α), and RARα/RXRα can bind and transactivate the rNtcp promoter, but not mNtcp or hNTCP, whereas hepatocyte nuclear factor 3 beta (HNF3β ie. FOXA2) mediates transcriptional repression of the Ntcp/NTCP promoter in all three species via directly binding to its response elements [17]. RARα regulates NTCP expression via upregulating promoter activity of the hNTCP gene, and thus supports HBV infection. Intriguingly, Ro41-5253 was identified to repress the hNTCP promoter by antagonizing RARα in a study screening for compounds that blocked hNTCP promoter activity. Similarly, CD 2665, a synthetic retinoid that inhibits RAR-mediated transcription, also downregulates NTCP expression. Glucocorticoid receptor (GR) could directly bind and activate NTCP promoter in a ligand-dependent manner. The hNTCP expression is upregulated by the GR ligand dexamethasone in Huh-7 cells and augmented by peroxisome proliferator-activated receptor-γ coactivator-1α (PGC-1α), while GR based activation is inhibited by FXR-induced expression of SHP [18]. The lack of peroxisome proliferator-activated receptor α (PPARα), a family of nuclear hormone receptors, increases NTCP mRNA in female mice via liver Fxr-Shp-Lrh-1 signaling. However, other studies have observed no effect of PPAR on NTCP promoter [65]. Additionally, members of the CCAAT/enhancer-binding protein (C/EBP) family contain two isoforms; the predominant basal C/EBP isoform in the liver is C/EBPα, whereas C/EBPβ is induced in the acute phase. Though they are isoforms, they have a different function in NTCP

expression. C/EBPα potentiates NTCP transactivation by retinoids, whereas C/EBPβ contributes to NTCP suppression [66].

STAT5, a member of the signal transducers and activators of the transcription family, activates the Ntcp promoter via binding to the interferon-gamma-activated sequence-like elements (GLEs) in the promoter, and mediates the upregulation of NTCP expression by prolactin. Interestingly, Ntcp expression is downregulated during pregnancy despite the increase in the level of STAT5 [64]. This may suggest that a STAT5-independent pathway activated during pregnancy is more important than the effect of STAT5 on Ntcp expression. Cyclin D1, a protein required for progression through the G1 phase of the cell cycle, can transcriptionally inhibit NTCP expression during cell cycle progression by inhibiting the activity of the NTCP promoter [63].

With regard to pro-inflammatory cytokines and endotoxin, almost all factors have negative effects on NTCP expression. IL-1β, a cytokine known to mediate acute phase changes in hepatic protein synthesis at the transcriptional level, reduces rNtcp and hNTCP promoter activity and transport activity by suppression of the RAR/RXR complex [15]. The effect of IL-1β on the RAR/RXR heterodimer was mediated by the c-Jun N-terminal kinase (JNK)-dependent pathway, but not extracellular signal-regulated protein kinase (ERK), and is abolished by curcumin, a JNK inhibitor [15]. Except for IL-1β, the other pro-inflammatory cytokines, such as TNF-α or IL-6, interfere with Ntcp/ NTCP expression at the mRNA and protein levels [67]. Oncostatin M (OSM), a member of the interleukin (IL)-6 family, mediates the downregulation of sinusoidal solute carrier (SLC) influx transporters, including NTCP [68].

With regard to chemical compounds, four compounds (dioxin, rifampicin, pheno-barbital, and oltipraz) have been shown to downregulate NTCP expression in PHHs and HepaRG cell lines. In addition to this, cholestyramine, which is a bile acid sequestrant in the intestine that increases bile acid biosynthesis in mouse liver, also increases mNtcp mRNA expression [61].

## 5. NTCP as a Target of Drug Development

HBV entry and bile acid transporters share common molecular determinants on NTCP [7], so it is crucial to study NTCP activity for the blockage of HBV infection. However, low uptake of bile acids by NTCP does not necessarily affect HBV entry [31], thus not all drugs that inhibit the uptake of bile acids can inhibit the entry of HBV.

### 5.1. Types of NTCP Inhibitors

There are two kinds of NTCP inhibitors. One can be transported by NTCP, while the other cannot. For example, NTCP substrates, including ursodeoxycholic acid, cholic acid, and taurocholate competitively inhibit NTCP-mediated bile acids. In addition to endogenous substrates, drugs transported by NTCP, such as statins (fluvastatin, pitavastatin, simvastatin, and rosuvastatin) [69], micafungin, CDCA-L-Val-ribavirin, and gadolinium-ethoxybenzyl-diethylenetriaminepenta-acetic acid can also be used as a competitive bile acid inhibitors.

Plenty of NTCP inhibitors have been reported, although they are not transported by NTCP, e.g., cholyl-L-lysyl-fluorescein (CLF) is a good inhibitor but not a substrate [70]. Peptidomimetic renin inhibitors propranolol, cyclosporin, oxysterols and progesterone, were found to be the potent inhibitors of NTCP [71]. Moreover, drugs such as bosentan, furosemide, probenecid, and bumetanide are also potent inhibitors of NTCP [10]. In 2012, scientists identified potential inhibitors via computational modeling of NTCP substrate specificity, and further screened 18 potential inhibitors from 37 drugs via their pharma-cophore model [72]. In another study, scientists identified thirty-one drug NTCP inhibitors while screening a total of 1280 drugs [21]. The list of NTCP inhibitors included thirty-six [20]. The renal tubular epithelial cell line (LLC-PK1) model, which stably expresses hNTCP, has been used for high throughput screening and utilized to verify the well-known cholestatic drugs, rifampicin, rifamycin SV, glibenclamide, and cyclosporine, as inhibitors

of NTCP. A similar study also identified six potential NTCP inhibitors from 102 herbal medicinal ingredients [73].

### 5.2. Development of Anti-HBV Drugs Targeting NTCP

Currently, numerous drugs have been developed for HBV treatment targeting NTCP. Initially, cyclosporin A (CsA) and its derivatives interrupt the binding of preS1 to NTCP to prevent HBV entry [74]. NTCP substrates, such as taurocholate, tauroursodeoxycholate, and bromosulfophthalein, also inhibited HBV infection [7,51,74]. Several HBV inhibitors that inhibit metabolic activity of NTCP, such as irbesartan, ritonavir, herbal medicines (vanitaracin A, proanthocyanidin and its analog oolong homobisflavan C), rosiglitazone, zafirlukast, TRIAC, sulfasalazine, and Chicago sky blue 6B, have been identified, but the precise mechanism has yet to be elucidated [75,76]. Interestingly, -(-)-epigallocatechin-3-gallate reduces HBV entry by accelerating the degradation of NTCP [77]. Furthermore, Ro41-5253 [78], IL-6 [16], and IL-1β [67] regulate the expression of NTCP, consequently raising the prospects of carrying out HBV treatment via downregulating of NTCP.

Recently, several drugs have been developed to block the interaction between HBV and NTCP. Bulevirtide (generic name, formerly known as myrcludex B), an HBV preS1-derived lipopeptide, is a competitive inhibitor of NTCP that can inhibit HBV infection effectively in humanized chimeric uPA mouse models [79], and has become the first approved treatment for adults with chronic HDV by the European Medicines Agency. The phase-III clinical trials at present have already proved the safety and efficacy profile of 2 mg dose once daily. In phase-IIb clinal trials, it was shown that there was a strong effect on HDV RNA levels in serum and induced ALT normalization. The combination of bulevirtide with PegIFNα-2a significantly reduced HBV DNA. Nevertheless, it had minimal effect on hepatitis B surface antigen [80]. The does-dependent efficacy of bulevirtide has also been found to improve biochemical activity and liver stiffness. Takaji Wakita's group synthesized two small macrocyclic peptides binding to NTCP, thus inhibiting HBV infection. Additionally, bile acid transport remained unchanged [81]. A list of anti-HBV drugs targeting NTCP has been summarized in Table 2.

**Table 2.** Drugs that target NTCP to treat HBV/HDV.

| Function | Drugs | Mechanism |
|---|---|---|
| Inhibit HBV/HDV infection | Proanthocyanidin [82] | Directly target the preS1 region of the HBV large surface protein |
| | Bile acids (taurocholate, tauroursodeoxycholate and bromosulfophthalein) [7,51,74] | Competition with preS1 |
| | Myrcludex B [80] | PreS1-derived lipopeptide, competition with preS1 |
| | WD1, WL2 [81] | Bind preS1 to inhibit HBV infection |
| | CsA and its derivatives [74] | Interrupt the binding of NTCP to PreS1 |
| | Irbesartan [75] | Interrupt NTCP function, be able to reduce HBeAg expression |
| | Ritonavir [76] | |
| | Vanitaracin A [83] | Directly interacts with NTCP and impairs its bile acid transport activity. |
| | Proanthocyanidin and its analogs Oolong homobisflavan C [82] | Targets amino acids 2–48 of the preS1 region and does not interfere NTCP-mediated bile acid transport activity |
| | Rosiglitazone, zafirlukast, TRIAC, sulfasalazine, and Chicago sky blue 6B [84] | NTCP inhibitor |
| | -(-)-Epigallocatechin-3-gallate [77] | Accelerates the degradation of NTCP |
| | IL-1β [67] | Activate the NF-κB signaling pathway |
| | TNF-α [67] | |
| | Ro41-5253 [78] | Downregulate NTCP expression |
| | IL-6 [16] | |

## 6. Conclusions

Chronic hepatitis is still regarded as the leading cause of liver cirrhosis and hepatocellular carcinoma worldwide. NTCP was originally discovered as a bile acid transporter in vivo. The identification of NTCP as the functional HBV and HDV receptor is a great breakthrough. This discovery not only opened new avenues in the field of drug discovery but also in developing models for screening anti-HBV drugs.

An ideal anti-HBV/HDV drug should inhibit HBV/HDV infection via NTCP but allow bile acid uptake. Screening FDA-approved drugs for their role in inhibiting HBV/HDV entry could be used as a starting point in developing novel therapies.

Successful clinical trial results for HBV entry inhibitor, bulevirtide, epitomizes the importance and forward-looking nature of entry inhibitors. Nonetheless, an eclectic approach is warranted, combining inhibition of viral entry and replication to treat chronic HBV patients [80].

In conclusion, NTCP is among the most critical factors in resisting the whole HBV infection process. In addition, there are other unknown host factors involving HBV infection with NTCP. The discovery of these host factors will greatly help us to generate HBV animal models (e.g., mouse HBV model) and facilitate antiviral drug development. Recently, glabridin, a natural product, inhibited HBV infection by enhancing the antiviral immune response as well as downregulating NTCP [85]. In the future, a more effective approach would be to develop drugs that could potentially inhibit HBV infection and activate innate immunity.

**Author Contributions:** Conceptualization, X.Z. (Xiaoling Zhou) and P.S.; writing, X.Z. (Xiaoyu Zhao), W.I., X.Z. (Xiaoling Zhou) and P.S; editing, W.I., X.Z. (Xiaoling Zhou) and P.S. All authors have read and agreed to the published version of the manuscript.

**Funding:** This work was funded by grants from National Natural Science Foundation of China, grant number 81870432 and 81570567 to X.L.Z.; 81571994 to P.N.S. and 81950410640 to W.I., the Natural Science Foundation of Guangdong Province, China (No. 2020A1515010054 to P.N.S.) and the Li Ka Shing Shantou University Foundation (Grant No. L11112008).

**Institutional Review Board Statement:** Not applicable.

**Informed Consent Statement:** Not applicable.

**Data Availability Statement:** Not applicable.

**Acknowledgments:** We would like to thank Stanley Lin for his guidance and insightful observation during the preparation of this manuscript.

**Conflicts of Interest:** The authors declare no conflict of interest.

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
