# Peer review of "Na+-Taurocholate Co-Transporting Polypeptide (NTCP) in Livers, Function, Expression Regulation, and Potential in Hepatitis B Treatment"

_livers, doi:10.3390/livers1040019_

Round 1

Reviewer 1 Report

Zhao et al. present a concise, well-formed review on NTCP. They cover both physiological function as well as its role in HBV/HDV infection.

1) Line 130: hNTCP expression has also shown HBV infection in rhesus macaques in vivo. This should be referenced.

2) Line 324: Myrcludex B actually has already received conditional approval for use in the EU under the tradename Hepcludex. This information should be included.

3) Information about antibody 2H5-A14 should be removed from the text and Table 2, since it has no relevance to NTCP targeting.

4) Line 352: This conclusion paragraph seems out of place and really doesn't summarize the manuscript as a whole. It should focus on NTCP. HBV avoiding innate immunity etc. really isn't on topic with a review about NTCP.

Author Response

Comments and Suggestions for Authors

Zhao et al. present a concise, well-formed review on NTCP. They cover both physiological function as well as its role in HBV/HDV infection.

1) Line 130: hNTCP expression has also shown HBV infection in rhesus macaques in vivo. This should be referenced.

Response: The authors have added the reference “Burwitz, B.J., et al., Hepatocytic expression of human sodium-taurocholate cotransporting polypeptide enables hepatitis B virus infection of macaques. Nat Commun, 2017. 8(1): p. 2146.” accordingly.

2) Line 324: Myrcludex B actually has already received conditional approval for use in the EU under the tradename Hepcludex. This information should be included.

Response: Changes were made based on the reviewer’s suggestion.

3) Information about antibody 2H5-A14 should be removed from the text and Table 2, since it has no relevance to NTCP targeting.

Response: The authors appreciate your suggestions and have revised it.

4) Line 352: This conclusion paragraph seems out of place and really doesn't summarize the manuscript as a whole. It should focus on NTCP. HBV avoiding innate immunity etc. really isn't on topic with a review about NTCP.

Response: We appreciate the reviewer’s comments and have revised it accordingly. We changed it into “In conclusion, NTCP is among the most critical factors in resisting the whole HBV infec-tion process. In additional, there are other unknown host factors involving HBV infection with NTCP. The discovery of these host factors will greatly help us to generate HBV ani-mal models (eg.mouse HBV model) and facilitate antiviral drug development. Recently, glabridin, a natural product, inhibited HBV infection by enhancing the anti-viral immune response as well as downregulating NTCP [88]. In the future, a more effective approach would be to develop drugs that could potentially inhibit HBV infection and activate innate immunity. “ (Refer to: Miyakawa, K., et al., Development of a cell-based assay to identify hepatitis B virus entry inhibitors targeting the sodium taurocholate cotransporting polypeptide. Oncotarget, 2018. 9(34): p. 23681-23694.)

Reviewer 2 Report

The authors reviewed the functions of NTCP including the association with hepatitis viruses. Although the review is interesting and informative for readers, several concerns are included as follows.

  1. Section 1 (Line 36-37): I think that ref No. 2 in which the association between HBV entry and NTCP was reviewed is not proper as the reference of this sentence. The authors should refer correctly.
  2. Section 3.2: The authors mentioned about species-specific liver tropism of HBV in this section. In my knowledge, NTCP expresses in the liver of all animals and can attach with HBV. However, HBV can invade to the inside of hepatocytes in the limited animal livers such as human, chimpanzee, and Tupaia, because of the lack of system for internalizing HBV in the animals which do not have the susceptibility of HBV. Therefore, the authors should explain more clearly.
  3. Section 3.2: How much amino acid sequences of NTCP are conserved among animals? Please explain this point in this section.
  4. Section 3.2: Are there any reports which describe about the association between HBV susceptibility and NTCP expression level in hepatocytes? If yes, please explain it.
  5. Section 3.2: Some variants of hNTCP such as A64T have been reported. Please discuss hNTCP variants more clearly including the differences of allele frequencies among human races and the impacts on HBV susceptibility.
  6. Section 3.3 (Line 176-178): Please explain the mechanism of the suppression of ISGs expression by bile acid transport clearly and indicate the proper references.
  7. Section 4 (Line 199-202): The authors mentioned the sexual difference of NTCP expression levels. Are there any impacts of this difference on HBV susceptibility? If yes, please explain in this section.
  8. Section 4: According to the previous report (Takeuchi JS, et al. Journal of Virology, 2018), NTCP variant G158R also related with the susceptibility of HBV. Please discuss about this variant.
  9. Section 5.1: Are there any impact of NTCP inhibitors on the metabolisms in human hepatocytes and the other organs? Please explain about the strong and weak points.
  10. Section 5.2 (Line 326): I think that Myrcludex B has been approved for the treatment to HDV infection in EU area and the phase 3 clinical study is undergoing in US. Please verify, and explain correctly.

Author Response

Comments and Suggestions for Authors

The authors reviewed the functions of NTCP including the association with hepatitis viruses. Although the review is interesting and informative for readers, several concerns are included as follows.

Section 1 (Line 36-37): I think that ref No. 2 in which the association between HBV entry and NTCP was reviewed is not proper as the reference of this sentence. The authors should refer correctly.

Response: We have deleted previous No.2 reference here and added the references “Ghany, M.G. and E.C. Doo, Antiviral resistance and hepatitis B therapy. Hepatology, 2009. 49(5 Suppl): p. S174-84.” and “Zoulim, F., Hepatitis B virus resistance to antiviral drugs: where are we going? Liver Int, 2011. 31 Suppl 1: p. 111-6.”

Section 3.2: The authors mentioned about species-specific liver tropism of HBV in this section. In my knowledge, NTCP expresses in the liver of all animals and can attach with HBV. However, HBV can invade to the inside of hepatocytes in the limited animal livers such as human, chimpanzee, and Tupaia, because of the lack of system for internalizing HBV in the animals which do not have the susceptibility of HBV. Therefore, the authors should explain more clearly.

Response: Many thanks for the reviewer’s suggestions. The reasons for species-specific liver tropism of HBV were summarized accordingly in this section and were closely related to NTCP variation, expression level or lack of system for internalizing HBV after NTCP attachment step.

NTCP is expressed in hepatocytes of all species, but not all animals NTCP can bind to HBV. The reason why some species can bind to HBV PreS1 is currently believed to be that amino acid No. 158 is glycine, while the NTCP sequence No. 158 of some non-susceptible cells is arginine.

We have added the explanation regarding to the internalizing system that the reviewer mentioned. The relevant explanation is clearly explained in the text.

Section 3.2: How much amino acid sequences of NTCP are conserved among animals? Please explain this point in this section.

Response: Thanks for the reviewer comment. We have added that amino acid sequence of NTCP is highly conserved in mammals (across 66.2% of the codons) (We added the reference:  40. Takeuchi, J.S., et al., A Single Adaptive Mutation in Sodium Taurocholate Cotransporting Polypeptide Induced by Hepadnaviruses Determines Virus Species Specificity. J Virol, 2019. 93(5).)

Section 3.2: Are there any reports which describe about the association between HBV susceptibility and NTCP expression level in hepatocytes? If yes, please explain it.

Response: Yes, hNTCP expression level is crucial for HBV infection. We explained in this section ” In vitro cell models, involving the HepaRG cell line [46], primary human hepatocytes (PHHs) [47] and primary Tupaia hepatocytes were found to be susceptible to HBV, whereas HepG2 and Huh-7 cell lines do not get infected due to low NTCP expression [25].” In addition, Reference showed that some drugs which can decrease NTCP expression can inhibit HBV infection (e.g.   Tsukuda, S., et al., Dysregulation of retinoic acid receptor diminishes hepatocyte permissiveness to hepatitis B virus infection through modulation of sodium taurocholate cotransporting polypeptide (NTCP) expression. J Biol Chem, 2015. 290(9): p. 5673-84.).

Section 3.2: Some variants of hNTCP such as A64T have been reported. Please discuss hNTCP variants more clearly including the differences of allele frequencies among human races and the impacts on HBV susceptibility.

Response: We appreciate the suggestions and have revised this section accordingly. Different variants have the different functional for inhibit HBV infection. Some variants, such as A64T and S267F have the low HBV infection efficiency because significantly decreased uptakes of taurocholate. And other variants, such as G158R can decrease the HBV infection with remain the transportation function at least 40%. With the similar effect, variants of NTCP does not support efficient taurocholate uptake and HBV entry, resulting in lower risk of chronic HBV infection. However, a controversial relationships still remain between S267F polymorphisms and HBV susceptibility.

We added the following information in this section. “With the discovery of NTCP as a key receptor for HBV infection, hNTCP residues 157 to 165 (KGIVISLVL) were identified to play an important role in this process.  [5]. By an-alyzing the susceptible of 5 NTCP variants (S142T, K157G, G158R, V160I, and L165P), Simon et.al have shown that amino acid position 158 of NTCP/Ntcp is critical for HBV/ HDV infection which 158G (glycine) is susceptible whereas 158R (Arginine) is non-susceptible [41]. But G158R also decrease the taurocholate uptake nearly 60%. Simi-larly, the substitution of serine at position 267 of NTCP with phenylalanine (S267F) [42]and the substitution of alanine at position 64 of NTCP with threonine (A64T) [43]previously were found to cause substantial loss of ability supported HBV and HDV infection and its bile acid uptake capability in vivo.

There are difference in allele frequencies of NTCP variants, such as race-specific var-iant S267F. It lose the function of transporting bile salts and the capability of binding HBV surface protein preS1, and thus blocking HBV entry. The allele frequencies of NTCP S267F ethically vary with 2.4-8.3% in Chineses [44], 7.5% in Chinese-Americans [45], 9.2% in Vi-etnamese and 3.1% in Koreans [43], while it was not found in European-Americans and African-Americans [45]. It was suspected that higher frequency of variant S267F in Asia population might resulted from NTCP gene evolution under a higher frequency of HBV infection[46]. However, a controversial relationships still remain between S267F poly-morphisms and HBV susceptibility [47-49]”

Section 3.3 (Line 176-178): Please explain the mechanism of the suppression of ISGs expression by bile acid transport clearly and indicate the proper references.

Response: We thank the reviewer for the comments and explained it in this section accordingly “Previously, it was discovered that bile acid has an inhibitory effect on the ISGs such as antiviral proteins MX1 and OAS1 in the interferon signaling pathway [58]. The HCV structural protein E2 does not bind to NTCP, in fact bile acids transported via NTCP sup-press the expression of antiviral ISGs, thus resulting in increased HCV infectivity.”.

Section 4 (Line 199-202): The authors mentioned the sexual difference of NTCP expression levels. Are there any impacts of this difference on HBV susceptibility? If yes, please explain in this section.

Response: It is reported that there was a significantly increased risk of HBV infections in men compared to women (Refer to: Ayano, G., et al., A systematic review and meta-analysis of gender difference in epidemiology of HIV, hepatitis B, and hepatitis C infections in people with severe mental illness. Ann Gen Psychiatry, 2018. 17: p. 16.). But whether it is related with the hNTCP expression in humans is not clear. Because although NTCP expression is higher in woman than in man, there is no significant difference between them. (Refer to: Cheng, X., D. Buckley, and C.D. Klaassen, Regulation of hepatic bile acid transporters Ntcp and Bsep expression. Biochem Pharmacol, 2007. 74(11): p. 1665-76.)

Section 4: According to the previous report (Takeuchi JS, et al. Journal of Virology, 2018), NTCP variant G158R also related with the susceptibility of HBV. Please discuss about this variant.

Response: Thanks for the reviewer’s useful suggestions. Changes are made based on reviewer’s suggestions.

Section 5.1: Are there any impact of NTCP inhibitors on the metabolisms in human hepatocytes and the other organs? Please explain about the strong and weak points.

Response: We divided the NTCP inhibitors into two kinds. One kind of inhibitors which inhibit NTCP physiological function and affect bile acid transport, may cause cholestasis or other disease. But the other one kind of inhibitors such as Hepcludex (Myrcludex B) can inhibit HBV infection without affect bile acid transport.

Section 5.2 (Line 326): I think that Myrcludex B has been approved for the treatment to HDV infection in EU area and the phase 3 clinical study is undergoing in US. Please verify, and explain correctly.

Response: We thank the reviewer for the comment. Changes have been made accordingly.

Round 2

Reviewer 2 Report

The authors answered to all the concerns raised by the reviewer. However, I think the third point which I raised at the first review has not been responded properly. Everyone knows that hNTCP expression is necessary for HBV infection and that HBV cannot infect with hepatocytes whose NTCP expression level is very low. Therefore, we want to know whether HBV is susceptible all hepatocytes which express NTCP with more than a certain level, or whether there are some reports which HBV susceptibility to human hepatocytes has an expression level dependency of hNTCP. Please explain this point in the section 3.2.

Author Response

Reviewer:

The authors answered to all the concerns raised by the reviewer. However, I think the third point which I raised at the first review has not been responded properly. Everyone knows that hNTCP expression is necessary for HBV infection and that HBV cannot infect with hepatocytes whose NTCP expression level is very low. Therefore, we want to know whether HBV is susceptible all hepatocytes which express NTCP with more than a certain level, or whether there are some reports which HBV susceptibility to human hepatocytes has an expression level dependency of hNTCP. Please explain this point in the section 3.2.

Response: HBV is not susceptible to all hNTCP-expressing hepatocytes (such as hNTCP-expressing mouse, rat, and dog hepatocytes cannot be infected by HBV), even though the hNTCP expression level is high. It was supposed that additional host factors are required for HBV infection besides hNTCP.  (Ref. Lempp, F.A., et al., Sodium taurocholate cotransporting polypeptide is the limiting host factor of hepatitis B virus infection in macaque and pig hepatocytes. Hepatology, 2017. 66(3): p. 703-716.)

Up to now, we cannot make conclusion that HBV susceptibility to human hepatocytes has an expression level dependency of hNTCP. hNTCP level and HBV infection rate may be not in parallel.

NTCP expression level of HepaRG-NTCP cells is higher than that of HepaRG cells and HBV infection rate of HepaRG-NTCP cells (~40%) is also higher than HepaRG cells (~20%). However, stem cell-derived hepatocytes expressed higher level of NTCP than human primary hepatocytes but the former had a lower HBV infection rate than the latter (Ref. Ni, Y. and S. Urban, Stem cell-derived hepatocytes: A promising novel tool to study hepatitis B virus infection. J Hepatol, 2017. 66(3): p. 473-475.).

In addition, Koichi Watashi’s group showed that different HepG2 clone isolates that similarly expressed high levels of ectopic NTCP, but had diverse efficiencies of HBV infection (Ref. Iwamoto, M., et al., Evaluation and identification of hepatitis B virus entry inhibitors using HepG2 cells overexpressing a membrane transporter NTCP. Biochem Biophys Res Commun, 2014. 443(3): p. 808-13. Watashi, K., et al., NTCP and beyond: opening the door to unveil hepatitis B virus entry. Int J Mol Sci, 2014. 15(2): p. 2892-905.)

We explained this point in the section 3.2 accordingly.